# Association between the rs3812316 Single Nucleotide Variant of the *MLXIPL* Gene and Alpha-Linolenic Acid Intake with Triglycerides in Mexican Mestizo Women

**DOI:** 10.3390/nu14224726

**Published:** 2022-11-09

**Authors:** Montserrat Maldonado-González, Zamira H. Hernández-Nazara, Nathaly Torres-Castillo, Erika Martínez-López, Lucia de la Cruz-Color, Bertha Ruíz-Madrigal

**Affiliations:** 1Laboratorio de Investigación en Microbiología, Departamento de Microbiología y Patología, Centro Universitario de Ciencias de la Salud, Universidad de Guadalajara, Guadalajara 44340, Jalisco, Mexico; 2Instituto de Investigación en Enfermedades Crónicas Degenerativas, Centro Universitario de Ciencias de la Salud, Universidad de Guadalajara, Guadalajara 44340, Jalisco, Mexico; 3Instituto de Nutrigenética y Nutrigenómica Traslacional, Centro Universitario de Ciencias de la Salud, Universidad de Guadalajara, Guadalajara 44100, Jalisco, Mexico; 4Centro de Investigación en Biotecnología Microbiana y Alimentaria, División de Desarrollo Biotecnológico, Centro Universitario de la Ciénega, Universidad de Guadalajara, Guadalajara 47820, Jalisco, Mexico

**Keywords:** G dyslipidemias, triglycerides, MLXIPL, ChREBP, alpha-linolenic acid, omega-3

## Abstract

The carbohydrate response element binding protein (ChREBP) is a key transcription factor to understand the gene–diet–nutrient relationship that leads to metabolic diseases. We aimed to analyze the association between the rs17145750 and rs3812316 SNVs (single nucleotide variants) of the *MLXIPL* gene with dietary, anthropometric, and biochemical variables in Mexican Mestizo subjects. This is a cross-sectional study of 587 individuals. Genotyping was performed by allelic discrimination. In addition, liver and adipose tissue biopsies were obtained from a subgroup of 24 subjects to analyze the expression of the *MLXIPL* gene. An in silico test of the protein stability and allelic imbalance showed that rs17145750 and rs3812316 showed a high rate of joint heritability in a highly conserved area. The G allele of rs3812316 was associated with lower triglyceride levels (OR = −0.070 ± 0.027, *p* < 0.011, 95% CI = −0.124 to −0.016), the production of an unstable protein (ΔΔG −0.83 kcal/mol), and probably lower tissue mRNA levels. In addition, we found independent factors that also influence triglyceride levels, such as insulin resistance, HDL-c, and dietary protein intake in women. Our data showed that the association of rs3812316 on triglycerides was only observed in patients with an inadequate alpha-linolenic acid intake (1.97 ± 0.03 vs. 2.11 ± 0.01 log mg/dL, *p* < 0.001).

## 1. Introduction

Over time, human beings have adapted to different environmental conditions, which has helped the survival and evolution of populations [1]. In addition, the abrupt changes in society have led to changes in diet and lifestyle, which, coupled with genetic susceptibility, play an important role in the development of non-communicable diseases. This is reflected in the presence of obesity and alterations in energy metabolism that in turn promote the development of the metabolic syndrome [2,3,4].

In Mexico, the most important contribution to energy intake comes from food products with a high content of saturated fats, trans fat, simple sugars, refined grains, and sodium, which have been associated with an increase in body weight, insulin resistance, and dyslipidemias, factors that affect the quality of life and the health of individuals [5,6,7].

Recently, studies have focused on the mechanisms of adaptation to deficit and excess of nutrients intake, which is crucial for maintaining metabolic homeostasis. Several genes that participate in mechanisms of regulation and storage of the energy from food have been identified [8]. The *MLXIPL* gene belongs to the MyC/Max/Mad family, and it is located in the long arm of chromosome 7q11.2. This gene codes the transcriptional factor ChREBP (Carbohydrate response element binding protein), which is activated by the intake of glucose and forms heterodimers with MLX (Max-like protein X) to carry out its transcriptional activity [9]. The ChREBP-MLXL complex recognizes a ChoRE (carbohydrate response element) consensus sequence to induce the transcription of genes involved in the glycolysis, lipogenesis, and gluconeogenesis pathways. *MLXIPL* is expressed mainly in the liver, white adipose tissue, small intestine, skeletal muscle, and pancreatic β cells [9]. There are two isoforms of this protein: ChREBP-α and ChREBP-β, out of which the latter possesses higher transcriptional activity [10]. Because ChREBP is activated by glucose, it is sensitive to changes in diet composition and regulates the expression levels of its two isoforms in a tissue-specific dependent manner [11].

GWAS (Genome-wide association study) studies have identified variants in the *MLXIPL* gene, notably the functional variant rs3812316 (Gln241His) located in an evolutionary conserved region. It encodes a domain involved in its activation, which has been associated with elevated triglycerides and decreased HDL-c in plasma, in addition to being proposed by several authors as a “thrifty gene” [8].

Other studies have identified that the single nucleotide variants (SNVs) rs3812316 and rs17145750 (C/T) are in high linkage disequilibrium in the Chinese population; in addition, they have been associated with a higher risk of cardiovascular disease, high triglyceride levels, and coronary heart disease, supporting the role of ChREBP in metabolic imbalance [12]. Nevertheless, other research has reported contradictory results [13,14,15], probably due to differences in the genetic background of different populations, as well as specific biological and environmental factors. To our knowledge, the effect on the structure and expression of the isoforms (ChREBPα and ChREBPβ) derived from the change of amino acids due to these SNVs is not fully understood. This study aims to analyze the association of the rs3812316 and rs17145750 SNVs in the *MLXIPL* gene with anthropometric factors, biochemical and dietary parameters, as well as the expression levels of the ChREBP-α and ChREBP-β isoforms in hepatic, omental, and subcutaneous adipose tissues, and its implication in the metabolism of a Mexican mestizo population.

## 2. Materials and Methods

### 2.1. Subjects

In this cross-sectional study, 587 subjects belonging to the Mexican Mestizo population genetically unrelated and randomly selected were voluntarily enrolled (Figure 1). The Mestizo population is defined as individuals born in Mexico, with a Spanish-derived last name, and a family ancestry of at least three generations [16]. It has been reported that, in Western Mexico, the paternal ancestry of Mexican-Mestizos is 60–64% [17]. This study was conducted at the Laboratorio de Investigación en Microbiología, Departamento de Microbiología y Patología, Centro Universitario de Ciencias de la Salud, Universidad de Guadalajara, from March 2019 to March 2021. The criteria to be included in the study considered age between 18–65 years old, indistinct sex, and no familiar relatedness; according to the BMI of participants, they were grouped as normoweight (18.5–24.9 kg/m^2^), overweight (25–29.9 kg/m^2^), and obesity (≥30 kg/m^2^). Subjects with the presence of chronic-degenerative or infectious disease, alcohol intake >20 g/day, pregnancy, consumption of glucose/lipid-lowering drugs, hormonal, and anti-inflammatory therapies were not included. Liver, omental (OAT), and subcutaneous adipose tissue (SAT) were obtained from a subgroup of 24 participants who underwent elective cholecystectomy surgery for reasons that do not interfere with the study. Biopsies were obtained from the Hospital Civil de Guadalajara.

### 2.2. Ethical Considerations

Subjects were informed about the procedures and informed consent was obtained from each participant (Appendix A). The study was approved by the Research Ethical Committee, CONBIOETICA-14CEI-002-20191003, Centro Universitario de Ciencias de la Salud, Universidad de Guadalajara. All procedures were conducted in accordance with The Code of Ethics of The World Medical Association based on the ethical guidelines of the 2013 Declaration of Helsinki.

### 2.3. Anthropometric Measurements

Body weight and body composition were determined using tetrapolar body electrical bioimpedance (InBody 3.70^®^. Biospace Co, Ltd., Seoul, Korea). The body mass index (BMI) was calculated as the weight in kilograms divided by the height in squared meters. The waist (the narrowest diameter between the lowest borders of the rib cage and the iliac crest) and hip (the widest portion of buttocks) circumferences were measured to calculate the waist–hip ratio (WHR) using a Lufkin Rosscraft^®^ tape (Lufkin Rosscraft^®^, Houston, TX, USA; model W606).

### 2.4. Nutritional Assessment

The subjects were asked to record their daily dietary intake for 3 days, including a weekend day. Participants were given prior written instructions to answer the dietary record; in addition, food replicas (Nasco^®^ Wisconsin, Fort Atkinson, WI, USA) were used to ensure a better understanding of food portion sizes. The records were reviewed by the same registered dietitian and analyzed with Nutritionist Pro™ Diet Analysis software (Axxya Systems, Stafford, TX, USA). Dietary data were handled with food scales and models based on Mexican food composition tables as a reference to enhance the accuracy of the portion size [18].

Diet Intake Definitions. Fat intake and specific fatty acid intake were categorized according to the Academy of Nutrition and Dietetics [19] as follows:Adequate total fat intake (Ad fat): 20–35% of kcal;Excessive total fat intake (Ex fat): >35% of kcal;Adequate total saturated fatty acid intake (Ad SFA): <7% of kcal;Excessive total saturated fatty acids intake (Ex SFA): >7% of kcal;Adequate total monounsaturated fatty acids intake (Ad MUFA): 15–25% of kcal;Insufficient total monounsaturated fatty acids intake (Ins MUFA): <15% of kcal;Adequate total polyunsaturated fatty acids intake (Ad PUFA): 3–10% of kcal;Insufficient total polyunsaturated fatty acids intake (Ins PUFA): <3% of kcal;Adequate total alpha-linolenic acid intake (Ad ALA): 0.6–1.2% of kcal;Insufficient total alpha-linolenic acid intake (Ins ALA): <0.6% of kcal.

### 2.5. Biochemical Measurements and Definitions

After 12 h of overnight fasting, venous blood samples were collected for biochemical determinations. The following parameters were tested: glucose, albumin, triglycerides, total cholesterol (TC), high-density lipoprotein cholesterol (HDL-c), and low-density lipoprotein cholesterol (LDL-c). The last one was calculated using the Friedewald’s formula excluding subjects with triglycerides over 400 mg/dL [20]. Very-low density lipoprotein cholesterol (VLDL-c) was calculated as total cholesterol minus (LDL-c + HDL-c), and non-HDL-cholesterol as total cholesterol minus HDL-c. Biochemical assays were simultaneously performed to minimize analytical variability on a dry chemistry system (Vitros 250 Analyzer, Ortho Clinical Diagnostics, Johnson & Johnson Co, Rochester, NY, USA). Insulin levels (Monobind Inc., Lake Forest, CA, USA) were measured to determine insulin resistance (IR) by the homeostasis model assessment (HOMA-IR) index as follows: HOMA-IR index = (fasting insulin [μU/mL] × fasting glucose [mg/dL]) divided by 405. HOMA-IR values over 2.5 were considered indicators of IR according to Matthews et al. [21].

Dyslipidemias were classified using the recommendations of the National Cholesterol Education Program Expert Panel on Detection, Evaluation, and Treatment of High Blood Cholesterol in Adults (NECP/ATP III) [22]. Hypertriglyceridemia was defined as fasting plasma concentration ≥ 150 mg/dL. Concentration of HDL-c <40 mg/dL in men or <50 mg/dL in women were considered abnormal, whereas high LDL-c levels were defined as >100 mg/dL. Finally, TC ≥ 200 mg/dL was considered hypercholesterolemia.

### 2.6. DNA Extraction and Genotyping

Genomic DNA (deoxyribonucleic acid) was extracted from leukocytes by a previously described method [23]. The rs17145750 and rs3812316 SNVs were determined by allelic discrimination with TaqMan^®^ Genotyping Assays (ID: C_2632498_10 and ID: C_33586301_10, respectively) in a LightCycler^®^ 96 Real-Time PCR System (Roche Diagnostics, Mannheim, Germany). The final reaction volume was 10 μL, which contained 2.5 μL of 20 ng/μL genomic DNA, 5 μL of 2× Fast Start Essential DNA Probes Master mix (Roche Diagnostics, Mannheim, Germany), 1 μL of 20× Genotyping Assay, and 1.5 μL nuclease-free water. Thermal cycling conditions were as follows: 95 °C for 10 min and 40 cycles of denaturation at 95 °C for 15 s and annealing/extension at 60 °C for 1 min. Reproducibility of the assays was verified using known genotype samples as positive controls and sterile distilled water as negative controls in each 96-well plate; 20% of the samples were performed in duplicate.

### 2.7. RNA Extraction of Tissue Samples

After collection, biopsies were cut (15 mg for hepatic and 100 mg for adipose tissues) and immediately processed for total RNA (ribonucleic acid) extraction using the TRIzol^®^ solution (ThermoFisher Scientific Inc., Waltham, MA, USA), and performed as previously described [24]. RNA quantity and purity were estimated by Nanophotometer P-Class (Implen Inc., Munich, Germany). The integrity of samples was checked by 1% agarose gel electrophoresis.

### 2.8. qPCR Real-Time

Reverse transcription was performed on 1 μg of high-quality RNA with a Transcriptor First Strand cDNA (complementary deoxyribonucleic acid) Synthesis Kit (Roche Molecular Biochemistry, Mannheim, Germany), following the instructions of the manufacturer. Samples were then amplified by qPCR (quantitative polymerase chain reaction) assays for the ChREBPα/β isoforms using UPL (universal probe library) probes (Roche, Mannheim, Germany) with the housekeeping gene RNA Pol II (ID: Hs999999901_s1, (TaqMan^®^), and conditions previously described [24]. All qPCR assays were performed in triplicate in a Light Cycler^®^ 96 System (Roche Diagnostics, Mannheim, Germany) according to the manufacturer’s instructions. The relative expression of the genes of interest was estimated using the equation 2^−ΔΔCq^.

### 2.9. In Silico Analysis of rs3812316 SNV of MLXIPL Gene

In silico analysis was performed in I-Mutant3.0 [25], which predicts the stability of mutated proteins based on the extent of changes in the amino acid structure. I-Mutant3.0 calculates the direction and ∆∆G (delta delta G) values considering thermodynamic data of mutated protein with a 77–80% of precision. A ∆∆G < 0 means a decrease in protein stability, whereas ∆∆G > 0 means an increase in protein stability.

### 2.10. Statistical Analyses

The sample size was calculated a priori considering the reported differences in triglyceride levels between the genotypes of the rs3812316 SNV [26] with an 80% statistical power, α = 0.05, and an effect size of 0.4; therefore, at least 84 individuals per group were required. Quantitative variables were expressed as mean ± standard deviation (SD) or median and interquartile range (IQR). The normality of quantitative variables was assessed with the Kolmogorov–Smirnov test. Comparison of quantitative variables between genotypes (dominant model) was performed using Student’s *t*-test or Mann–Whitney U test as appropriate. To compare the frequencies of qualitative variables, as well as Hardy–Weinberg equilibrium, the chi-square test was used. To estimate the haplotype frequency for both SNVs, the EM algorithm for haplotype inference with multiallelic markers was used. The linkage disequilibrium (D′) and squared correlation of allele frequencies (r2) were calculated with the SHEsis online software [27]. The association of the SNV with dyslipidemias was assessed by binary logistic regression adjusting for confounders. Gen-nutrient association on triglyceride levels was analyzed with the ANCOVA (analysis of covariance) test using Bonferroni correction for multiple comparisons. In ANCOVA analysis, the classification of fat and specific fatty acid intake (according to the position of the Academy of Nutrition and Dietetics) was employed. The in silico analysis was performed with the I-Mutant (version 3.0) software [25] using the FASTA protein sequence as reported in UniProt [28]. All statistical analyses were computed in SPSS version 20.0 (IBM Corp., Armonk, NY, USA). The sample size was calculated with the software G*Power version 3.1.9.7 [29]. Graphics were created with GraphPad Prism version 8.3.1 (GraphPad Software, San Diego, CA, USA) and with BioRender.com (accessed on 14 September 2022).

## 3. Results

### 3.1. Characteristics of the Study Population

In this study, a total of 587 subjects were included, out of which 70.6% were women with a mean age of 40.9 ± 11.9 years. According to BMI classification, 20.8% were normal-weight, 30.5% were overweight, and 48.7% were subjects with obesity. Women and men differ in all anthropometric values, insulin, HOMA-IR, HDL-c, albumin, energy intake, and kilocalories consumed from carbohydrates, protein, and fat, as well as some fatty acids and fiber. These differences are shown in Table 1, Table 2 and Appendix A. The baseline characteristics for the cholecystectomy subgroup are reported in Appendix A.

### 3.2. Frequencies of SNVs rs17145750 and rs3812316 of MLXIPL Gene

The genotype frequencies of rs17145750 (C/T) and rs3812316 (C/G) SNVs in all participants are shown in Table 3. The genotype frequencies of both SNVs were in Hardy Weinberg equilibrium and no differences were observed in genotype or allelic frequencies between women and men. Likewise, both SNVs were in linkage disequilibrium (D′ = 0.840, r^2^ = 0.533).

### 3.3. Anthropometric, Biochemical, and Nutritional Variables by rs17145750 and rs3812316 SNVs

Due to the differences found in metabolic, anthropometric, and nutritional characteristics between men and women, we decided to carry out the subsequent analysis by sex. Regarding the rs17145750 SNV, it was found that women with the CC genotype had a higher intake of MUFA 22:1 and PUFA 22:5. Other variables in men and women, including triglyceride levels, did not show significant differences (Figure 2A and Appendix A). Concerning the rs3812316 C > G genotype, women with the CG + GG genotypes showed lower serum triglyceride levels than those with the CC genotype (Figure 2B). Other comparisons are reported in Appendix A.

### 3.4. Association of rs17145750 and rs3812316 SNV’s of MLXIPL Gene with Dyslipidemias

The distribution of dyslipidemia frequencies was analyzed between genotypes of rs17145750 C > T and rs3812316 C > G (Appendix A). In women, a higher prevalence of hypertriglyceridemia was found in those with the CC genotype compared to the CG + GG genotypes (38.1 vs. 19.0%, *p* = 0.005) of the rs3812316 SNV. No differences were found in men in either of the two SNVs.

Based on these results, we compared all anthropometric, biochemical, and nutritional variables in subjects with and without hypertriglyceridemia. Those variables that displayed a significant difference were introduced in a multiple logistic regression analysis to know which were associated with hypertriglyceridemia. In women, waist circumference, total cholesterol, and glucose were risk factors for hypertriglyceridemia; meanwhile, HDL-c and the rs3812316 SNV were protective factors (Figure 3A). In men, risk factors were fat mass and TC, whereas protein intake and HDL-c were protective factors (Figure 3B). The B coefficients, 95% Confidence Interval, and *p*-value of the logistic regression models are available in Appendix A.

### 3.5. Association of rs3812316 SNV of MLXIPL Gene with Triglyceride Levels

Because we found differences in the triglyceride levels according to rs3812316 genotypes, we analyzed which other anthropometric, biochemical, and nutritional variables were correlated to triglycerides. Such variables were introduced in a multiple linear model with the stepwise method (see all models in Appendix A) to describe how much they contribute to the variability in triglycerides concentration. In women, HDL-c and the SNV rs3812316 C > G were associated with lower triglyceride levels, whereas total cholesterol, waist circumference, and HOMA-IR were associated with higher triglyceride levels (Table 4). In men, HDL-c and dietary fiber were related to lower triglycerides; on the contrary, body fat percentage, albumin, and total cholesterol were related to higher triglyceride concentrations (Table 4).

### 3.6. SNVs and Nutrient Association

Since fat intake is one of the main nutrients that influence triglyceride levels, we studied a possible association between fat intake and specific types of fat with the SNV rs3812316. For this purpose, the ANCOVA analysis was used, adjusted for the variables that were previously associated with triglycerides. Data showed that the concentration of triglycerides was lower in individuals with the G allele, regardless of the type of fat consumption. Women with the CC genotype, regardless of excessive or adequate consumption of total fat or SFA, had higher triglyceride levels than those with the CG + GG genotype; in addition, those with the CC and an insufficient intake of MUFA or PUFA had higher triglycerides than those with the CG + GG genotypes. Subject carriers of the CC genotype with adequate intake of MUFA or PUFA had lower triglyceride levels compared to those with insufficient diet in the same lipids, without reaching statistical significance. However, the lowest levels of triglycerides were found in individuals with insufficient diet intake of MUFA or PUFA and the CG + GG genotype.

Interestingly, triglyceride levels were elevated in women with insufficient intake of alpha-linolenic acid (18:3 n-3) and were higher in the CC carriers than in those with the CG + GG genotypes. There was no significant difference in the levels of serum triglycerides between individuals with CC and CG + GG genotypes and an adequate intake of alpha-linolenic acid (Figure 4). In men, none of the nutrients showed a significant association with the SNV (data not shown).

### 3.7. Expression of ChREBP α and β Isoforms by Genotype

In a sub-group of 24 participants with overweight and obesity, we examined the expression of ChREBP-α and ChREBP-β in the liver, omental, and subcutaneous adipose tissue according to the rs17145750 and the rs3812316 SNVs. Expression of ChREBP-α was lower in subjects with the CG genotypes in the liver and omental tissues, whereas an expression of ChREBP-β of CG carriers was reduced in omental and subcutaneous tissues; nevertheless, no significance value was detected (Figure 5A,B). The heterozygotes participants for the rs3812316 SNV were also heterozygotes for the rs17145750. Therefore, the same results were obtained for both SNVs, which in turn showed strong linkage disequilibrium (D’ = 1.0, r^2^ = 1.0).

### 3.8. In Silico Analysis of the SNV rs3812316

Finally, considering all the results obtained in this study regarding the rs3812316 C > G, we performed an in silico analysis to test the effect of amino acid change in the SNV. It was found that the change from glutamine to histidine diminished the ∆∆G value to −0.83 kcal/mol, which predicts a decrease in protein stability.

## 4. Discussion

In this study, we analyze the association of the rs3812316 and rs17145750 SNVs of the *MLXIPL* gene in a Mexican mestizo population. We found differences in clinical and nutritional characteristics by sex. To our knowledge, this is the first study that analyzes such SNVs along with anthropometric, biochemical, and dietary parameters, as well as their expression according to the rs3812316 and the rs17145750 genotypes in hepatic and adipose tissues in subjects without manifestation of chronic diseases.

The participants of this study showed significant differences between men and women in the anthropometric, biochemical, and nutritional variables. The majority of these variables were higher in men than women. Men had a higher energy intake than women. Other authors have reported similar results [2]. In contrast, women had a higher percentage of body fat.

On the other hand, when the individuals were classified by sex to analyze the distribution of the rs3812316 and rs17145750 SNVs, we identified that most of the significant associations were found in female carriers of the CC genotype of the rs3812316 SNV, whose serum levels of triglycerides and VLDL-c were higher than the carriers of CG/GG genotypes. In addition, we found an association between the consumption of fat and types of fat with the rs3812316 SNV. No differences or associations were found in men, probably because the posteriori statistical power was 20% in men and 80% in women.

The C allele of rs3812316 has been proposed to allow for a more efficient conversion of excess carbohydrates into fatty acids, which leads to greater use of the energy obtained and an increase in adipose tissue that would serve as a defense mechanism in prolonged fasting periods [8]; however, in a condition of overfeeding, it may lead towards the development of metabolic diseases, which are reported as a leading cause of mortality [2,3,4]. The association of elevated triglyceride levels with the rs3812316 SNV has been identified by other association studies of the *MLXIPL* gene in Mexican, Asian, and Caucasian populations [8,26,30,31]. Other studies carried out in the Chinese population tried to identify an association between different SNVs of the *MLXIPL* gene, but only reported an association of the rs3812316 with the risk of cardiovascular disease [12]. However, other studies have failed to find associations of these variables with the rs3812316 SNV [13,14,15].

The Mexican population is characterized by a high prevalence of hypertriglyceridemia and low HDL-c levels. The combination of these lipid traits is known as atherogenic dyslipidemia, which affects 19.5% of the population in Mexico [32,33]. The imbalance between energy intake and energy expenditure due to food consumption habits is a leading factor related to these lipid alterations, which, when coupled with genetic predisposition, may lead to pathophysiological events independently of other metabolic risks.

We analyzed the frequency of isolated dyslipidemias in this population to know the association between the SNVs and other environmental factors. We found that high triglyceride levels in women were associated with higher levels of total cholesterol, higher glucose, greater waist circumference, and as protective factors, higher serum levels of HDL-c and the presence of the G allele of the rs3812316. These models were corroborated with a linear regression where HOMA-RI replaced glucose. Studies in patients diagnosed with cardiovascular disease have also reported that carriers of the minor G allele have lower triglyceride concentrations and attributed a protective effect of this allele to the development of early atherosclerosis. In addition, murine models suggest that it induces a decrease in the function of ChREBP [12,34].

The amount of protein in diet plays a protective factor against hypertriglyceridemia in men, as it has been reported that subjects with a lower protein intake (0.8 g/kg of total body weight) consume more carbohydrates and added sugars [35], which in turn favors elevated triglyceride levels [36]. On the contrary, fiber, niacin, and omega-3, such as alpha-linolenic acid, decrease them. Several studies have confirmed that diets high in protein have positive effects on metabolic parameters, including lower triglyceride levels [37]. However, the restriction of protein also decreases triglyceride levels by reducing the rate of secretion of VLDL-c particles from the liver towards peripheral tissues, as well as a probable increase in the expression of apoA5 and fatty acid oxidation, which stimulates VLDL-c clearance [38]. Therefore, further studies are required to clarify this association.

We also found a gene–nutrient association: when the intake of MUFAs or certain PUFAs was low, carriers of the G allele are predicted to have lower triglyceride levels than carriers of the CC genotype. In line with our results, Ortega-Azorín and cols. reported that, among subjects whose food intake adhered to the Mediterranean diet, those with the minor allelic frequency (MAF) polymorphic G allele showed a lower risk of hypertriglyceridemia [26]. Several studies had demonstrated that monounsaturated fatty acids reduce triglyceride levels [39,40,41,42], partly because there is an increase in the secretion of apo E and apo C-III-rich VLDL and IDL particles, which in turn, reduces the time of VLDL in circulation; in contrast, lipoproteins without apo E or apo C-III have been shown to decrease lipolysis and clearance, which leads to its metabolization into LDL [43]. Studies in mice have shown that PUFAs can decrease ChREBP activity by altering its translocation to the nucleus due to a decline in xylulose 5-phosphate concentration, which is an activator of ChREBP [44]. Some authors have suggested that the G allele reduces the activity of ChREBP [8]; in support of this hypothesis, the expression of hepatic lipogenic enzymes of ob/ob-Chrebp−/− mice was lower than ob/ob mice, which leads to the normalization of triglyceride levels [45]. This scenario, in which the absence of ChREBP leads to lower triglyceride concentrations, is similar to that in which the presence of the genetic variant of rs3812316 leads to lower activity or expression of ChREBP.

The effect of the G allele in the reduction of triglycerides serum levels was affected by dietary fat intake, namely adequate intake of MUFAs, PUFAs, and alpha-linolenic acid. Particularly, the difference in triglyceride serum levels related to the rs3812316 SNV loses its significance with an adequate intake of alpha-linolenic acid. This phenomenon is explained by the fact that the adequate intake of alpha-linolenic acid modifies the lipidome [46]; in addition to the esterification in the carbon skeleton of the glycerol that constitutes the triglycerides, alpha-linolenic acid is also relevant because it is a constituent of other phospholipid membrane components such as phosphatidylcholine. Modern Western dietary changes promote diets insufficient in PUFAs of the omega-3 series. Furthermore, alpha-linolenic acid is a precursor of anti-inflammatory and pro-resolution lipid mediators such as docosahexaenoic acid (DHA) and eicosapentaenoic acid (EPA); therefore, it prevents metabolic diseases [47].

In the present study, we analyzed the influence of the SNV rs3812316 on the mRNA expression of ChREBP isoforms in a subgroup of 24 subjects with overweight or obesity in different tissues. Different patterns of expression levels were observed for the isoforms of the *MLXIPL* gene. Regarding the α isoform, the CG genotype of rs3812316 presented lower levels of expression in liver and omental adipose tissue (not significant) compared to the wild-type CC genotype. The lack of significance is probably due to the low frequency of the G allele carriers and the null presence of GG homozygotes. In contrast, the expression of ChREBP-β in liver tissue showed similar levels of expression when compared to the CG genotype; however, this genotype shows a statistical tendency towards a lower expression of this isoform in adipose tissue (omental and subcutaneous) compared to the genotype CC. The gene expression analysis of the rs17145750 SNV was the same as rs3812316 because both SNVs were in strong linkage disequilibrium, as previously reported [12]. It has been observed that a decrease in the expression of ChREBP-β in white adipose tissue reduces glucose uptake and de novo lipogenesis, factors associated with an improvement in insulin sensitivity [10,35]. Therefore, the reduced expression of ChREBP-β in white adipose tissue supports the importance of the G allele of rs3812316 as a protector factor. The change in the amino acid sequence provoked by the SNV rs3812316 (Gln241His) is located in the evolutionarily conserved GRACE domain, which is responsible for the response to glucose by the *MLXIPL* gene [48].

In silico analysis of the sequence with the rs3812316 SNV of the *MLXIPL* gene showed that the presence of the G allele leads to a decrease in protein stability. The change from the wild C allele to the G allele yields a value of ∆∆G −0.83 kcal/mol, which indicates a dysfunctional folding of the ChREBP protein, a destabilization in its structure that may lead to reduced levels of the protein. Furthermore, it has been speculated that the allelic change (C > G) found in GRACE may lead to lower affinity and reduced activity [4], thus explaining the lower levels of mRNA expression of *MLXIPL* in G carriers compared to C carriers. In addition, the in silico analysis shows a probable disruption in the protein structure due to the allelic change, which could lead to a decrease in the post-transcriptional activity of ChREBP in carriers of the G allele.

This study has some limitations, such as a lower proportion of men than women. In addition, diet intake was evaluated through diet records that possess a margin of uncertainty. Furthermore, the small number of sample biopsies did not provide enough statistical power to detect differences, in case they exist. Therefore, although one of our strengths was to measure the expression of ChREBP according to genotypes in the liver and adipose tissues, this comparison needs to be made in a larger number of samples. In addition, it is necessary to replicate this study in Mexican and other populations, as well as to analyze SNVs in other genes related to triglyceride metabolism such as APOE, MTTP, and CEPT, and to describe their effect on the response to a dietary intervention.

## 5. Conclusions

In conclusion, this study demonstrated that genetic factors such as the rs3812316 of *MLXIPL*, biological factors such as high glucose levels, high cholesterol, low HDL-c, and elevated waist circumference, as well as behavioral factors such as low protein and fiber intake, are related to triglyceride levels in Mexican women. In addition, we found a gene-nutrient diet association, especially related to fat and types of fat intake, such as alpha-linolenic acid. Finally, our data showed that the association of the rs3812316 on triglycerides was only observed in subjects with inadequate alpha-linolenic acid intake.

## Figures and Tables

**Figure 1 nutrients-14-04726-f001:**
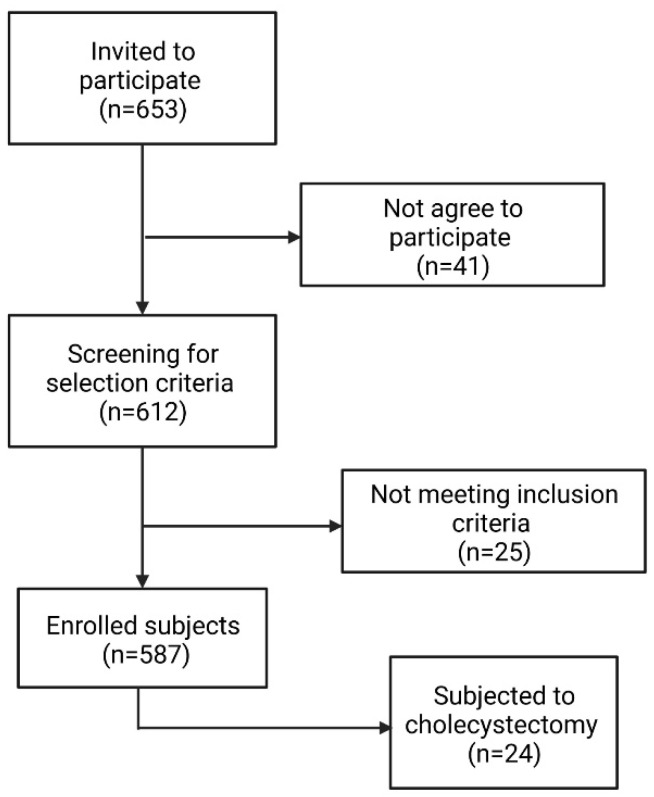
Flowchart for recruitment of the subjects.

**Figure 2 nutrients-14-04726-f002:**
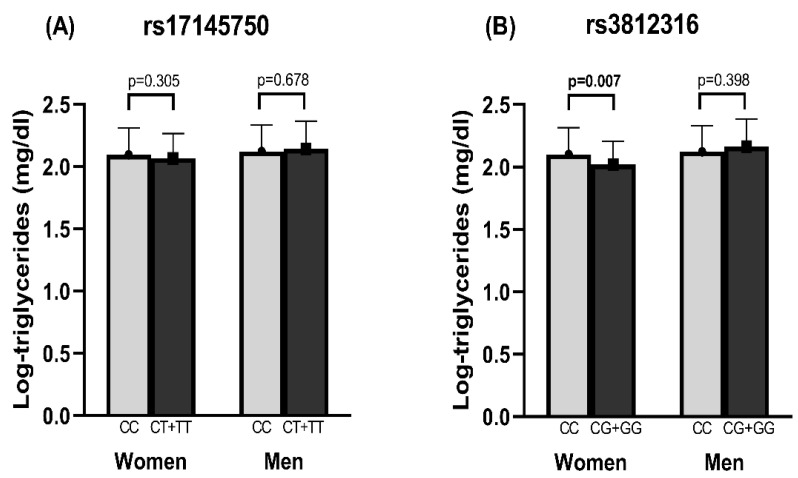
Triglyceride levels according to SNV’s in *MLXIPL* gene. Comparison of triglyceride levels in women and men according to (**A**) rs17145750 and (**B**) rs3812316 SNVs. Bars represent mean ± SD (standard deviation). Triglyceride concentrations were log10-transformed to ensure a normal distribution and comparisons between women and men were performed with Student’s *t*-test. A *p*-value < 0.05 was considered statistically significant. Bold numbers highlight statistical significance.

**Figure 3 nutrients-14-04726-f003:**
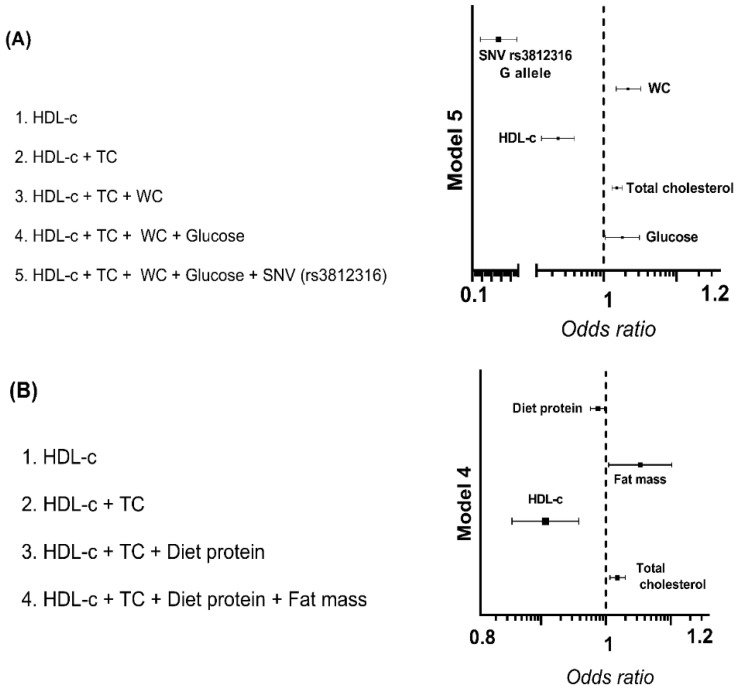
Factors associated with hypertriglyceridemia in women and men. Logistic regression models of hypertriglyceridemia in (**A**) women and (**B**) men. Binary logistic regression was performed with hypertriglyceridemia (triglycerides ≥ 150 mg/dL) as a dependent variable. Lines represent the Odds Ratio and the 95% Confidence Interval. A *p*-value < 0.05 was considered statistically significant. HDL-c: High-density lipoprotein cholesterol, TC: Total cholesterol, WC: Waist circumference, SNV: Single nucleotide variant.

**Figure 4 nutrients-14-04726-f004:**
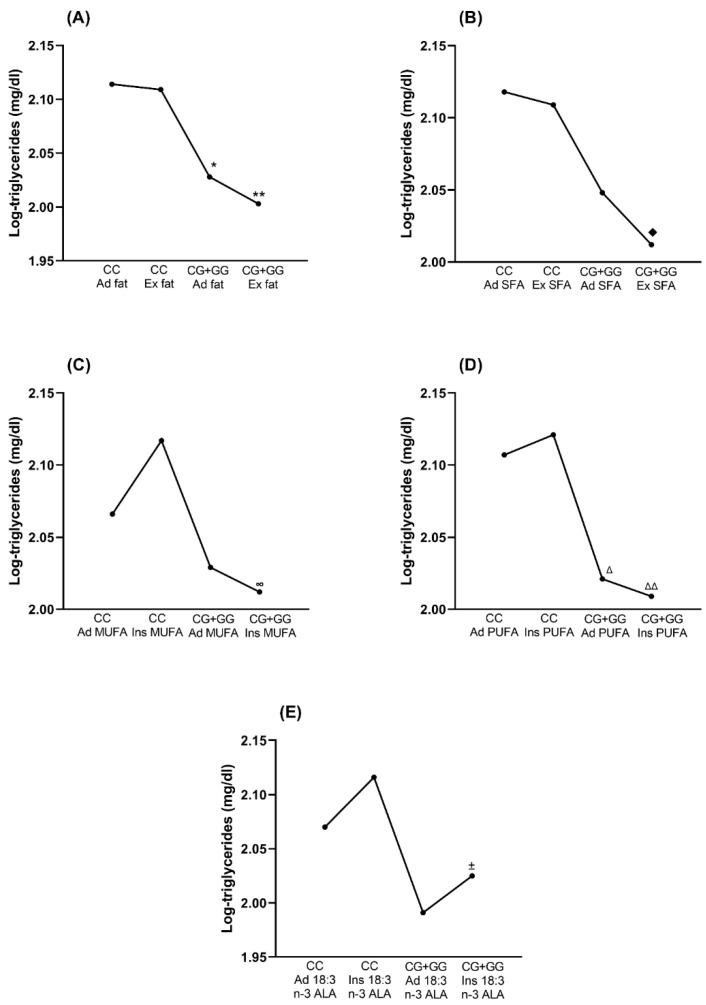
Triglyceride levels according to the rs3812316 C > G SNV and its association with (**A**) Total fat intake, (**B**) Saturated fatty acids, (**C**) Monounsaturated fatty acids, (**D**) Polyunsaturated fatty acids, and (**E**) alpha-linolenic acid in women. Data are shown as estimated mean. Fat intake and specific fatty acid intake were categorized according to the position of the Academy of Nutrition and Dietetics [19]. Triglycerides were log10-transformed to apply ANCOVA (analysis of covariance) analysis, adjusted for HDL-c, total cholesterol, waist circumference, HOMA-IR, and energy intake; Bonferroni correction was used to test for multiple comparisons. A *p*-value < 0.05 was considered statistically significant. SFA: Saturated fatty acids, MUFA: Monounsaturated fatty acids, PUFA: Polyunsaturated fatty acids, ALA: alpha-linolenic acid, n-3: omega-3, HDL-c: High-density lipoprotein cholesterol, HOMA-IR: Homeostasis model assessment of insulin resistance, Ad: adequate, Ex: excessive, Ins: insufficient. * CC adequate fat vs. CG + GG adequate total fat, *p* < 0.05. ** CC excessive fat vs. CG + GG excessive fat, *p* < 0.05. ♦ CC excessive SFA vs. CG + GG excessive SFA, *p* < 0.05. ∞ CC insufficient MUFA vs. CG + GG insufficient MUFA, *p* < 0.05. Δ CC adequate PUFA vs. CG + GG adequate PUFA, *p* < 0.05. ΔΔ CC insufficient PUFA vs. CG + GG insufficient PUFA, *p* < 0.05. ± CC insufficient ALA vs. CG + GG insufficient ALA, *p* < 0.05.

**Figure 5 nutrients-14-04726-f005:**
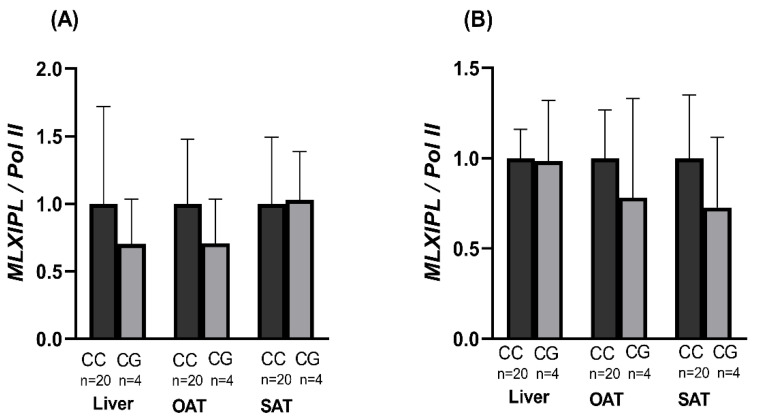
Gene expression of ChREBP α and β isoforms according to the rs3812316 SNV of MLXIPL gene. Units of relative expression (**A**) of ChREBP-α and (**B**) ChREBP-β isoforms according to the rs3812316. Bars represent mean ± SD (standard deviation). Comparisons between genotypes were performed with the Mann–Whitney U test. A *p*-value < 0.05 was considered statistically significant. Gene expression was performed by qPCR and using the 2^−∆∆CT^ method. OAT: Omental adipose tissue, SAT: Subcutaneous adipose tissue, qPCR: quantitative polymerase chain reaction.

**Table 1 nutrients-14-04726-t001:** Characteristics of the study population.

Variable	Total *n* = 587	Women *n* = 415	Men *n* = 172	*p*-Value
Age (y) Age range	40.93 ± 11.86 18–65	41.69 ± 11.71 18–65	39.14 ± 12.04 18–65	**0.001**
Anthropometric variables				
BMI (kg/m^2^)	30.78 ± 6.77	31.38 ± 6.88	29.36 ± 6.29	**0.001**
Waist circumference (cm)	96.12 ± 16.38	95.23 ± 16.09	98.21 ± 16.93	**0.043**
Hip circumference (cm)	109.44 ± 13.63	111.24 ± 14.17	105.17 ± 11.22	**<0.001**
WHR	0.86 ± 0.08	0.85 ± 0.08	0.91 ± 0.08	**<0.001**
Fat mass (g)	31.1 ± 14.57	33.49 ± 13.93	25.62 ± 14.56	**<0.001**
Body fat percentage (%)	36.55 ± 10.67	40.29 ± 8.71	27.77 ± 9.65	**<0.001**
Biochemical variables				
Glucose (mg/dL)	93.96 ± 18.22	93.34 ± 16.8	95.44 ± 21.21	0.196
Insulin (μU/mL)	12.74 ± 12.81	13.6 ± 12.86	10.6 ± 12.47	**0.001**
HOMA-IR	2.99 ± 3.26	3.19 ± 3.36	2.48 ± 2.94	**0.001**
Triglycerides (mg/dL)	141.54 ± 72.54	138.56 ± 73.34	148.62 ± 70.3	0.075
TC (mg/dL)	187.64 ± 41.47	187.73 ± 42.32	187.42 ± 39.5	0.935
VLDL-c (mg/dL)	28.3 ± 14.5	27.71 ± 14.66	29.72 ± 14.06	0.123
LDL-c (mg/dL)	115.91 ± 37.67	115.65 ± 38.69	116.5 ± 35.23	0.803
HDL-c (mg/dL)	43.42 ± 13.6	44.36 ± 14	41.19 ± 12.35	**0.010**
Non-HDL-c (mg/dL)	144.22 ± 41.87	143.37 ± 42.39	146.23 ± 40.68	0.448
Albumin (g/dL)	4.29 ± 0.48	4.19 ± 0.43	4.52 ± 0.50	**<0.001**

Data are presented as mean ± SD (standard deviation). Comparisons between women and men were performed with Student’s *t*-test. A *p*-value < 0.05 was considered statistically significant. Bold numbers highlight statistical significance. BMI: Body mass index, WHR: Waist-to-hip ratio, HOMA-IR: Homeostasis model assessment of insulin resistance, TC: Total cholesterol, VLDL-c: Very low-density lipoprotein cholesterol, LDL-c: Low-density lipoprotein cholesterol, HDL: High-density lipoprotein cholesterol.

**Table 2 nutrients-14-04726-t002:** Nutritional characteristics of the study population.

Variable	Total *n* = 587	Women *n* = 415	Men *n* = 172	*p*-Value
energy (kcal)	2025.0 ± 842.2	1866.7 ± 715.2	2390.4 ± 989.5	**<0.001**
Carbohydrates (kcal)	1000.6 ± 436.5	923.5 ± 377.9	1178.5 ± 506.4	**<0.001**
Carbohydrates (%)	49.68 ± 9.32	49.76 ± 9.29	49.51 ± 9.40	0.768
Proteins (kcal)	350.73 ± 146.99	322.27 ± 124.29	416.58 ± 172.58	**<0.001**
Proteins (%)	17.81 ± 4.56	17.74 ± 4.73	17.98 ± 4.15	0.570
Fat (kcal)	672.8 ± 361.58	619.83 ± 317.46	795.37 ± 423.58	**<0.001**
Fat (%)	32.44 ± 8.68	32.41 ± 8.74	32.50 ± 8.56	0.903
Saturated fat (g)	25.03 ± 14.78	22.98 ± 12.96	29.76 ± 17.45	**<0.001**
Monounsaturated fat (g)	22.89 ± 13.23	21.03 ± 11.40	27.18 ± 15.95	**<0.001**
Polyunsaturated fat (g)	11.9 ± 8.62	11.4 ± 7.94	13.06 ± 9.95	**0.034**
SFA 6:0 (g)	0.26 ± 0.28	0.24 ± 0.25	0.31 ± 0.33	**0.025**
SFA 8:0 (g)	0.22 ± 0.23	0.2 ± 0.21	0.26 ± 0.27	**0.018**
SFA 10:0 (g)	0.38 ± 0.36	0.35 ± 0.33	0.44 ± 0.40	**0.021**
SFA 14:0 (g)	1.87 ± 1.50	1.73 ± 1.38	2.19 ± 1.72	**0.002**
SFA 16:0 (g)	11.91 ± 7.19	10.91 ± 6.18	14.21 ± 8.70	**<0.001**
SFA 18:0 (g)	5.19 ± 3.40	4.75 ± 2.97	6.21 ± 4.05	**<0.001**
MUFA 14:1 (g)	0.06 ± 0.08	0.06 ± 0.07	0.08 ± 0.09	**0.006**
MUFA 16:1 (g)	1.14 ± 0.88	1.04 ± 0.72	1.39 ± 1.12	**<0.001**
MUFA 18:1 (g)	20.05 ± 12.04	18.37 ± 10.25	23.94 ± 14.71	**<0.001**
PUFA 18:2 (g)	9.96 ± 7.83	9.53 ± 7.08	10.94 ± 9.27	**0.049**
PUFA 18:3 (g)	0.95 ± 0.87	0.9 ± 0.93	1.04 ± 0.72	0.087
PUFA 20:4 (g)	0.14 ± 0.12	0.13 ± 0.11	0.16 ± 0.13	**0.005**
Total n-3 (g)	1.05 ± 0.97	1.01 ± 1.05	1.15 ± 0.75	0.114
Total n-6 (g)	10.1 ± 7.86	9.66 ± 7.11	11.1 ± 9.3	**0.045**
n-3:n-6 ratio	11.05 ± 7.13	11.03 ± 6.36	11.1 ± 8.68	0.916
Dietary cholesterol (mg)	293.08 ± 212.82	268.72 ± 182.81	349.44 ± 261.8	**<0.001**
Dietary fiber (g)	25.74 ± 13.27	24.1 ± 12.09	29.52 ± 15.03	**<0.001**
Soluble fiber (g)	0.53 ± 0.86	0.46 ± 0.79	0.68 ± 0.98	**0.008**
Insoluble fiber (g)	1.18 ± 1.95	1.01 ± 1.79	1.58 ± 2.22	**0.004**
Crude fiber (g)	4.73 ± 13.24	4.44 ± 12.91	5.4 ± 13.98	0.426
Total sugar (g)	76.67 ± 42.02	73.34 ± 40.04	84.47 ± 45.49	**0.006**
Added sugars (g)	21.02 ± 19.49	21.71 ± 20.84	19.49 ± 16.08	0.329
Glucose (g)	8.08 ± 7.25	8.02 ± 7.07	8.23 ± 7.65	0.753
Galactose (g)	0.42 ± 3.57	0.42 ± 3.68	0.42 ± 3.29	0.998
Fructose (g)	10.82 ± 10.35	10.56 ± 9.76	11.43 ± 11.61	0.393
Sucrose (g)	19.32 ± 19.85	19.51 ± 21.13	18.9 ± 16.58	0.740
Lactose (g)	10.7 ± 12.26	9.25 ± 10.45	14 ± 15.16	**<0.001**

Data are presented as mean ± SD (standard deviation). Comparisons between women and men were performed with Student’s *t*-test. A *p*-value < 0.05 was considered statistically significant. Bold numbers highlight statistical significance. SFA: Total saturated fatty acids, MUFA: Total monounsaturated fatty acids, PUFA: Total polyunsaturated fatty acids.

**Table 3 nutrients-14-04726-t003:** Genotype and allelic frequencies of the rs17145750 and rs3812316 SNV’s of *MLXIPL* gene.

	Total *n* = 587 *n* (Proportion)	Women *n* = 415 *n* (Proportion)	Men *n* = 172 *n* (Proportion)	*p*-Value
**SNV rs17145750 C > T**				
**Genotype**				
CC	490 (0.834)	342 (0.823)	148 (0.861)	0.329
CT	92 (0.157)	71 (0.170)	22 (0.127)	0.215
TT	5 (0.009)	2 (0.007)	2 (0.012)	0.585
p (HWE)	0.767	0.407	0.266	
**Allele**				
C	1072 (0.91)	755 (0.91)	318 (0.92)	0.493
T	102 (0.09)	75 (0.09)	26 (0.08)	0.493
**Dominant model**				
CC	490 (0.834)	342 (0.823)	148 (0.861)	
CT + TT	97 (0.166)	73 (0.177)	24 (0.139)	
**SNV rs3812316 C > G**				
**Genotype**				
CC	508 (0.865)	357 (0.861)	151 (0.878)	0.598
CG	78 (0.133)	57 (0.137)	21 (0.122)	0.689
GG	1 (0.002)	1 (0.002)	0 (0)	1.000
p (HWE)	0.262	0.414	0.393	
**Allele**				
C	1094 (0.93)	771 (0.93)	323 (0.94)	0.611
G	80 (0.07)	59 (0.07)	21 (0.06)	0.611
**Dominant model**				
CC	508 (0.865)	357 (0.860)	151 (0.878)	
CG + GG	79 (0.135)	58 (0.140)	21 (0.122)	

Data are presented as “*n*” and proportions. Comparison of frequencies between women and men was performed with the X^2^ test. A *p*-value < 0.05 was considered statistically significant. *p*-value of genotype and allelic frequencies comparisons between women and men. p(HWE): *p*-value of Hardy Weinberg equilibrium, SNV: single nucleotide polymorphism.

**Table 4 nutrients-14-04726-t004:** Parameters associated with triglyceride levels in women and men.

	B	Standard Error	95% CI	*p*-Value
*WOMEN*				
*Model,* R^2^ = 65.2				
HDL-cholesterol (mg/dL)	−0.005	0.001	−0.007–−0.004	**<0.001**
Total cholesterol (mg/dL)	0.002	0.000	0.001–0.002	**<0.001**
Waist circumference (cm)	0.003	0.001	0.002–0.004	**<0.001**
SNV rs3812316 C > G	−0.070	0.027	−0.124–−0.016	**0.011**
HOMA-IR	0.007	0.003	0.001–0.013	**0.015**
*MEN*				
*Model,* R^2^ = 79.4				
HDL-cholesterol (mg/dL)	−0.007	0.001	−0.010–−0.004	**<0.001**
Percentage of body fat (%)	0.007	0.002	0.003–0.010	**<0.001**
Albumin (g)	0.108	0.032	0.044–0.171	**0.001**
Total dietary fiber (g)	−0.004	0.001	−0.007–−0.001	**0.005**
Total cholesterol (mg/dL)	0.001	0.000	0.000–0.002	**0.032**

The linear regression models were performed with triglycerides transformed to log10 as a dependent variable. A *p*-value < 0.05 was considered statistically significant. Bold numbers highlight statistical significance. HDL-cholesterol: High-density lipoprotein cholesterol, HOMA-IR: Homeostasis model assessment of insulin resistance, CI: Confidence interval. B: is the beta coefficient for lineal regression and represents the change in the dependent variable for every unit of change in the independent variable. R^2^: is expressed in percentage and represents the percentage of variance in triglyceride levels explained by the model.

## Data Availability

The data presented in this study are available on request from the corresponding author.

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
