# Peer review of "Association between the rs3812316 Single Nucleotide Variant of the MLXIPL Gene and Alpha-Linolenic Acid Intake with Triglycerides in Mexican Mestizo Women"

_nutrients, 2022, doi:10.3390/nu14224726_

Round 1

Reviewer 1 Report

This is a very interesting work. The authors are kindly invited to revisit the following parts of the manuscript.

Lines 40, 42, 51, 54: Please provide relevant references to support the statements.

Line 78-86: Please remove the template instructions 

Kindly consider improving the introduction and briefly explain why this work is relevant. Also please try to maintain a targeted tone on the interactions that the authors have found in previous research (literature) and those that the authors aim to investigate in this work. It is essential for the reader to have a clear view of what the research gap is and how this work aims to tackle it.

Line 90: Please specify if the participants were voluntarily enrolled. Also, perhaps a table of inclusion/exclusion criteria would be beneficial for the reader. Kindly consider including (if applicable) the age limit established in the protocol. 

Please consider briefly explaining how much of the overall population in Mexico the Mestizo population represents.

Kindly consider providing a blank informed consent template. It is essential for the reader to have a clear understanding that the participants had been informed and had access to all necessary information (including anonymity coding and future use of the records/blood/biopsy) before their enrollment in the study.

Line 119: Please briefly explain if the participants were provided with guidelines/training / or any other form of help before they were asked to complete the 3d intake report. 

Line 134: Please explain when and how instructions were given to participants before blood samples were taken. Also briefly explain if dropouts were present and how they were handled (if applicable). 

Lines 220-223: Please remove the template instructions

Table 1: Please consider providing an age range for the participants.

Table 4: Please consider briefly explaining the R2 and B values presented in the table. Is this the regression fit? 

Line 379: Please discuss if this finding is dissimilar to previous reports.

Line 449: Since the classification is relevant to the analysis of the results, perhaps the authors would like to consider providing information on how the participants are stratified in the various BMI groups (in Table 1).

Please revisit the discussion segment of the manuscript and if possible try to maintain coherence in the presentation of comparisons with previous findings in the field.

Reviewer 2 Report

The manuscript shows the results of a cross-sectional study on 587 mixed Mexicans and a subset of 24 obese individuals from which histological samples were obtained. The work aimed to evaluate the SNP effect of the MLXIPL gene on metabolic parameters and nutritional aspects.

The authors' evaluations revealed an association between variants of rs3812316 and triglyceride levels in women. In addition, an association has emerged with the intake of fatty acids, especially between unsaturated fatty acids and triglyceride levels associated with the variants of rs3812316 but only in women. The authors proposed a mechanism of the action thanks also to the in silico analysis of the stability of the ChREBP product based on the polymorphism studied. No significant difference, however, in the case of gene expression analyzed by quantitative rt-PCR of the mRNA extracted from tissue samples in the subgroup of 24 individuals.

The topic is very interesting, the study is well described and the supporting literature is adequate. The organization of the manuscript is well structured and the language is fluent and understandable.

There are, however, some aspects that deserve consideration:

- The most important thing is the title, in my opinion not very representative. The authors found that triglyceride levels are lower in the gene variant of rs3812316 in case of insufficient intake of ALA. Therefore it is incorrect to say that the intake of ALA reduces the risk of hypertriglyceridemia associated with the SNV studied.

I suggest an alternative title more consistent with the study design and the results: "Association between the rs3812316 single nucleotide variant of the MLXIPL gene and triglycerides among Mexican Mestizo women"

- The abstract is very confusing and should be simplified by explaining the main features of the study and the main results. The use of a structured abstract is strongly recommended

- In the abstract and other parts of the manuscript, given the nature of the study, I recommend avoiding talking about interaction and using "association"

- Specify in both the manuscript and the methods that histological specimens are derived from a subset of 24 obese individuals. In addition, a table with baseline characteristics and significant differences from the whole group should be added to the results

- The dynamics of recruitment are not clear (except for the subgroup subjected to cholecystectomy). How were the participants recruited? How many have been proposed to participate? How many have accepted? How many were eligible for the study? Is the number of participants representative of the population group studied? A flow chart would elegantly solve this aspect.

- Among the limitations of the study it should be noted that nutritional habits have been extrapolated from questionnaires so they may have a margin of uncertainty. Furthermore, the power analysis was evaluated for the gene variants and not for the gene expression in the subgroup

- Each acronym should be defined in full on first use, as in the case of GWAS

- On line 68, perhaps the authors are referring to rs17145750.

- Parts of the journal template persisted in the manuscript such as lines 78-86 and 220-223

- In the caption of tables 1 and 2, reference is made to the text in bold in case of significance which, however, has not been used

- Instead, in figure 1 they refer to bold in case of significance even if no numeric data are shown

- In line 250, specify that the differences were found in the metabolic, anthropometric and nutritional characteristics

- Standardize the acronyms MUFA and PUFA with the MFA and PFA used in the tables

- Figure 2 has a scarce resolution

Round 2

Reviewer 2 Report

No further comments. Thank you